# Direct CD32 T-cell cytotoxicity: implications for breast cancer prognosis and treatment

Giuseppe Sconocchia[1], Giulia Lanzilli[1], Valeriana Cesarini[1], Domenico A Silvestris[2], Katayoun Rezvani[3], Roberto Arriga[4], Sara Caratelli[1], Ken Chen[3], Jinzhuang Dou[3], Carlo Cenciarelli[1], Gabriele Toietta[5], Silvia Baldari[5], Tommaso Sconocchia[6], Francesca De Paolis[1], Anna Aureli[1], Giandomenica Iezzi[7], Maria Irno Consalvo[8], Francesco Buccisano[8], Maria I del Principe[8], Luca Maurillo[8], Adriano Venditti[8], Alessio Ottaviani[1,*], Giulio C Spagnoli[1,*]

The FcγRII (CD32) ligands are IgFc fragments and pentraxins. The existence of additional ligands is unknown. We engineered T cells with human chimeric receptors resulting from the fusion between CD32 extracellular portion and transmembrane CD8α linked to CD28/ζ chain intracellular moiety (CD32-CR). Transduced T cells recognized three breast cancer (BC) and one colon cancer cell line among 15 tested in the absence of targeting antibodies. Sensitive BC cell conjugation with CD32-CR T cells induced CD32 polarization and down-regulation, CD107a release, mutual elimination, and proinflammatory cytokine production unaffected by human IgGs but enhanced by cetuximab. CD32-CR T cells protected immunodeficient mice from subcutaneous growth of MDA-MB-468 BC cells. RNAseq analysis identified a 42 gene fingerprint predicting BC cell sensitivity and favorable outcomes in advanced BC. ICAM1 was a major regulator of CD32-CR T cell–mediated cytotoxicity. CD32-CR T cells may help identify cell surface CD32 ligand(s) and novel prognostically relevant transcriptomic signatures and develop innovative BC treatments.

## Introduction

Innate immune cells represent the first line of defense against pathogens, virally infected, and cancer cells. Innate immune cells comprise a variety of myeloid and lymphoid cell (ILC) types. Myeloid cells include granulocytes, monocyte/macrophages, and DCs (Bardoel et al, 2014; Italiani & Boraschi, 2014). These cells regulate the host's inflammatory responses by releasing cytokines (Sconocchia et al, 2001; Suzuki et al, 2008), performing phagocytosis (Italiani & Boraschi, 2014), mediating direct cytotoxic functions (Pericle et al, 1996), and tissue repair following ischemic and thermal injuries (Roos et al, 2004; Gravante et al, 2007, 2009). Lymphoid cells include NK cells and ILC mainly involved in tissue homeostasis and mucosal immunity. Among innate immune cell receptors, Fc γ receptors (FcγRs) play key roles in the regulation of NK, ILC (Simoni & Newell, 2018), DC, and myelomonocytic cell functions (Simoni & Newell, 2018). By mediating antibody-dependent cellular cytotoxicity (ADCC), phagocytosis (ADP), and cytokine release, they are critically involved in the response to infectious challenges and the pathogenesis of inflammatory, and infectious diseases, as well as in tissue damage and remodeling (Pericle et al, 1996; Lanier, 2000; Sconocchia et al, 2001; Nimmerjahn & Ravetch, 2008; Bottazzi et al, 2016; Patel et al, 2019).

FcγRs include three distinct sets of molecules: the activating FcγRI (CD64), FcγRII (CD32, CD32C), and FcγRIII (CD16A and CD16B), and the single inhibitory FcγRII (CD32B). FcγRs bind γ immunoglobulins (Igs) with different affinities (Clynes et al, 1998; Nimmerjahn & Ravetch, 2006, 2008; Rosales, 2017). Whereas CD64 is a high-affinity receptor expressed on IFNγ-activated granulocytes (Sconocchia et al, 2001), monocyte/macrophages, and DCs (van der Poel et al, 2011), CD32 and CD16 are polymorphic with low binding affinity for monomeric IgG.

Fcγ activating receptors transduce signals through the immunoreceptor tyrosine-based activation motif (ITAM). CD64 and CD16A consist of dimers composed of a ligand-binding chain and a signal subunit γ chain containing ITAM. Instead, CD32 and CD32C are monomeric directly transducing activation signals through the ITAM located in the intracellular tail (Nimmerjahn & Ravetch, 2008).

[1]Department of Biomedicine Institute of Translational Pharmacology (IFT), National Research Council (CNR), Rome, Italy    [2]Department of Oncohematology, IRCCS Ospedale Pediatrico "Bambino Gesù", Rome, Italy    [3]Department of Stem Cell Transplantation and Cellular Therapy, Division of Cancer Medicine, MD Anderson Cancer Center, University of Texas, Houston, TX, USA    [4]Department of Systems Medicine, the University of Rome "Tor Vergata", Rome, Italy    [5]Tumor Immunology and Immunotherapy Unit, IRCCS Regina Elena National Cancer Institute, Rome, Italy    [6]Department of Internal Medicine, Division of Hematology, Medical University of Graz, Graz, Austria    [7]Department of Surgery, Università Svizzera Italiana, Lugano, Switzerland    [8]Department of Biomedicine and Prevention, University of Rome "Tor Vergata", Rome, Italy

Correspondence: giuseppe.sconocchia@ift.cnr.it
Roberto Arriga's present address is Takis Biotech, Rome, Italy.
*Alessio Ottaviani and Giulio C Spagnoli contributed equally to this work.

CD16A is the most important FcγR mediating ADCC. It is expressed in NK cells and small subsets of monocytes and NK T cells (Ziegler-Heitbrock et al, 1993; Sconocchia et al, 2005; Simoni & Newell, 2018). NK cells efficiently mediate ADCC against tumor cells. However, despite the high expression of NK cell ligands including MICA/B, in the microenvironment of solid cancers, NK cell infiltration is rarely detectable and has no direct prognostic impact (Sconocchia et al, 2009, 2011, 2012, 2014b; Coppola et al, 2015). In contrast, CD8+ T cells are promptly identified in the solid tumor microenvironment. These cells are associated with a favorable clinical course of the disease and are easy to expand in vitro.

Following the demonstration that rituximab could redirect CD16A-engineered T cells to kill B lymphoblastic cells (Clemenceau et al, 2006), we and others have independently produced a variety of CD16A chimeric receptors (CR) T cells to target solid tumor cells (Kudo et al, 2014; Ochi et al, 2014; D'Aloia et al, 2016). Also, NK cells have demonstrated weak ADCC, when used in combination with IgG2 mAb including panitumumab, whereas IgG2 mAb elicits ADCC in CD32 positive myeloid cells (Schneider-Merck et al, 2010). Therefore, to overcome the CD16A limitation, human CD32-CR (hereafter referred to as CD32-CR) has also been produced (D'Aloia et al, 2016; Caratelli et al, 2017, 2020; Arriga et al, 2020). They share with a classical CAR the intracellular tail, resulting from the fusion of the human transmembrane CD8α with the co-stimulatory molecule CD28 linked to the human T-cell CD3ζ chain (CD32/CD8a/CD28/ζ). In contrast, the extracellular CAR single-chain variable fragment (scFv), which recognizes the target tumor-associated antigen (TAA), has been replaced with the human extracellular portion of CD32. CD32-CR T cells bind IgG2 (panitumumab) even in the presence of human serum, whereas human CD16A-CR does not. In addition, panitumumab successfully redirects CD32-CR T cells toward cancer cells overexpressing EGFR (Arriga et al, 2020; Caratelli et al, 2020; D'Aloia et al, 2016).

Other than Igs, FcγRs bind pentraxins, a family of soluble proteins involved in opsonization, phagocytosis, and complement activation. The best known pentraxins are the C reactive protein and serum amyloid component (SAP) (Bharadwaj et al, 1999, 2001). Most interestingly, however, previous evidence suggests that at least CD16A mediates NK cell cytotoxicity in the absence of cell surface antigen-bound antibodies consistent with its ability to directly recognize poorly defined, cell surface ligand(s) expressed by tumor target cells (Mandelboim et al, 1999).

To date, the existence of human CD32 cell surface putative ligand(s) is still unknown. In this study, using CD32-CR as a biosensor, we provide evidence of the existence of CD32 surface ligand(s) on cancer cells and that CD32-CR T cells can be used as a therapeutic and prognostic tool for early clinical studies. The presented work is likely to enhance the basic and translational field of CD32 investigation behind ADCC and ADP.

# Results

### Molecular and phenotypic characterization of CD32-CR T cells

Following a 3-d stimulation, PBMCs were transduced with CD32-CR as described in the methods section. On day 14, non-transduced (NT) and transduced (T) cells underwent cell surface phenotypic analysis. About one hundred percent of NT and T cells were CD3+. The transduction efficiency was 80%. Bulk NT T cells and CD32-CAR T cells were collected for single-cell RNA sequencing (scRNA seq). Upon completion of quality filtering, a total of 1,613 cells were retained from four products for subsequent analysis. Data visualization by uniform manifold approximation and projection (UMAP) showed cell population differences between two products (Fig 1A). To further evaluate transcriptomic profile differences between NT T cells and CD32-CAR T cells, we identified up-regulated differentially expressed genes (DEGs) and the corresponding gene enriched pathways at product levels (Fig 1B and C). We found that NT T cells comprised mostly of hallmark pathways associated with, fatty acid metabolism and glycolysis as well as mTOR and c-Myc signaling, whereas CD32-CAR T cells showed an increased enrichment in interferon response pathway and TNFα, consistent with activation of pro-inflammatory signaling. Again, using cell function scores based on specified gene sets, transduced T cells displayed a significantly higher inflammatory and pro-apoptotic activation pathway than NT T cells (Fig 1D). To further characterize CD32-CR T cells, we performed a phenotypic analysis of three healthy donors by 12-color flow cytometry. The percentage of CD32 engineered CD8+ T cells was predominant over CD4+ T cells ([88.43 ± 4.6 versus 11.9 ± 4.6] P = 0.0003). Based on this result, we focused on CD3+CD32+CD8+ T cells. Fig 2 shows representative data from one donor. CD45RA expression identified the RA+ subset as significantly overrepresented than RA− (P = 0.05) and both RA+ and RA− were CD95+. Then, CCR7 and CD62L expression, within the CD3+CD32+CD8+CD45RA+CD95+CD27± cells, led to the recognition of four cell subsets (Fig 2, lower middle panels). Two of them, e.g. CD62L−CCR7− and CD62+CCR7+, could be identified as T effector memory RA ($T_{EMRA}$) and T stem cell memory ($T_{SCM}$), respectively. In contrast, neither the CD62L+CCR7− nor the CD62L−CCR7+ populations could be precisely classified. All four cell subsets included a relevant population of CD69+ cells, some of which expressed PD1 suggesting a sustained activation state. Similarly, to RA+, RA− cells were CD3+CD32+CD8+CD95+CD27±. However, these cell subsets were mainly $T_{EM}$ (CD62L−CCR7−) and to a minimal extent $T_M$ (CD62L−CCR7$^{low}$CD27+).

### CD32-CR promotes specific T-cell surface recognition of BC cells

To provide evidence of the existence of putative CD32 cell surface ligand(s) on cancer cells, we tested the ability of tumor cells from 15 established cell lines to trigger CD32-CR T cells (Table S1). Incubation with triple-negative breast cancer (TNBC) cells, MDA-MB-468 and MDA-MB-231, induced significant CD32-CR down-regulation in 78.7% ± 7% and 81.3% ± 8% of engineered T cells, respectively (Fig 3A). Concomitantly, CD107a mobilization was also observed. Representative contours and cumulative data are reported in Fig 3A. To provide visualization of the effects elicited by the interaction between BC and engineered T cells, we performed a confocal microscopy analysis. CD32-CR T cells (Fig 3B, panel a) and MDA-MB-468 (Fig 3B, panel b) were indirectly stained with anti-CD32 mAb (green) and anti-EGFR mAb (red), respectively. Incubation of CD32-CR T cells with MDA-MB-468 cells resulted in the formation of lytic synapsis including firm adhesion (Fig 3B panel c), CD32-CR polarization and down-regulation (Fig 3B panels d and e), and structural cell aberrations (Fig 3B panel f) including cell blebbing formation in

**Life Science Alliance**

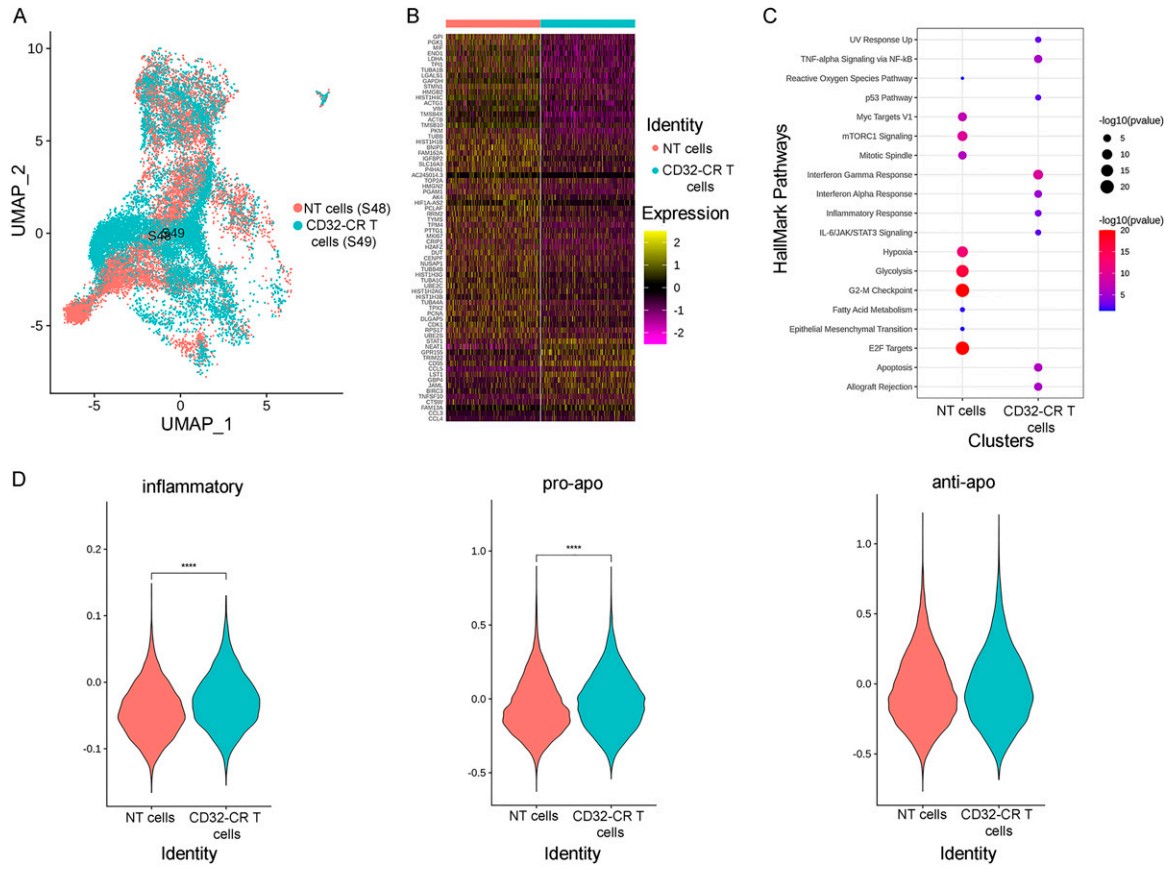

**Figure 1.  Single-cell gene expression analysis of CD32-CR T cells.**
**(A)** UMAP plots of CD32-CR transduced (T) and non-transduced (NT) T cells (NT = 11,504 cells, T = 9,123 cells). Each dot represents the transcriptome of a single cell, with color-coding transduced and non-transduced T cells. **(B)** Gene-expression heat map of the 89 differentially expressed genes in the CD32-CR T versus NT T cells (see the Materials and Methods section). Genes are represented in rows and single-cell expression levels in columns. Colors reflect the scaled gene expression levels from low (blue) to high (red). **(C)** Hallmark pathways enrichment based on up-regulated expressed genes across NT T and CD32-CR T-clustered cells (see the Materials and Methods section). Rows depict pathways, whereas columns depict samples. Both the size and color of any point correspond to $-\log_{10}$ transformed adjusted $P$-values from Fisher exact tests. Significant pathways were identified at an adjusted $P$-value < 0.01. **(D)** Shows differential function states between NT and transduced T cells in one of two representative healthy donors: proapoptotic (left) apoptotic (middle) and inflammatory (right) pathways are indicated (see the Materials and Methods section), ****$P$ < 0.0001 (Fisher exact test).

MDA-MB-468 cells (red blebs) and MDA-MB-468 and CD32-CR T-cell membrane fusion (yellow blebs). These results indicate that CD32-CR senses natural ligand(s) on TNBC cells.

### CD32-CR mediates a direct T-cell anti-cancer activity

To assess whether CD32-CR triggering might result in direct anti-tumor activity, MDA-MB-468[LUC] (luciferase-expressing) cells were incubated in the presence or absence of CD32-CR T cells, whereas NT T cells were used as negative controls. After 72-h incubation at 37°C, engineered T cells significantly reduced the biosignal produced by MDA-MB-468[LUC] cells even at a very low E:T ratio (0.125:1), whereas the NT T cells were completely ineffective (Fig 3C).

Because human IgG can bind CD32-CR (Caratelli et al, 2020), binding of CD32-CR to its putative ligand(s) may be hindered in the presence of serum Igs. Therefore, MDA-MB-468 cells were incubated for 72 h, at 37°C, in the presence or absence of NT T or engineered T cells with or without human serum (Fig 3D), or in a complete medium supplemented with scalar concentrations of human IgGs (Fig 3E).

Interestingly, CD32-CR T cell anti-cancer activity was neither affected by human AB serum (10–30% concentration), as shown by the reduced crystal violet staining of adherent MDA-MB-468 cells (Fig 3D), nor by human IgGs (Fig 3E). Similar results were obtained when the FcR blocking buffer was used. Also, the contribution of pentraxins to the direct anti-tumor activity of CD32-CR T cells was assessed. Both C reactive protein and SAP failed to protect the MDA-MB-468 viability from CD32-CR T cells (Fig S1). These results strongly suggest that the direct anti-tumor effect of CD32-CR is IgG-and pentraxin-independent and that the known and putative ligands have distinct binding sites.

To demonstrate that the direct anti-TNBC activity of CD32-CR T cells implies T cell–mediated cytotoxicity, CD32-CR T cells were incubated for 3 h with MDA-MB-468 cells at a 1:1 E:T ratio. Engineered T cells induced direct apoptosis/necrosis of MDA-MB-468 cells (Fig 4A, lower left panel), whereas NT T cells did not (Fig 4A, middle-left panel). In addition, the cytotoxic activity of cetuximab (Fig 4A, upper right panel) was minimally affected by NT T cells (Fig 4A, middle-right panel), whereas it was significantly enhanced by CD32-CR T cells (Fig 4A, lower right panel). These data indicate that

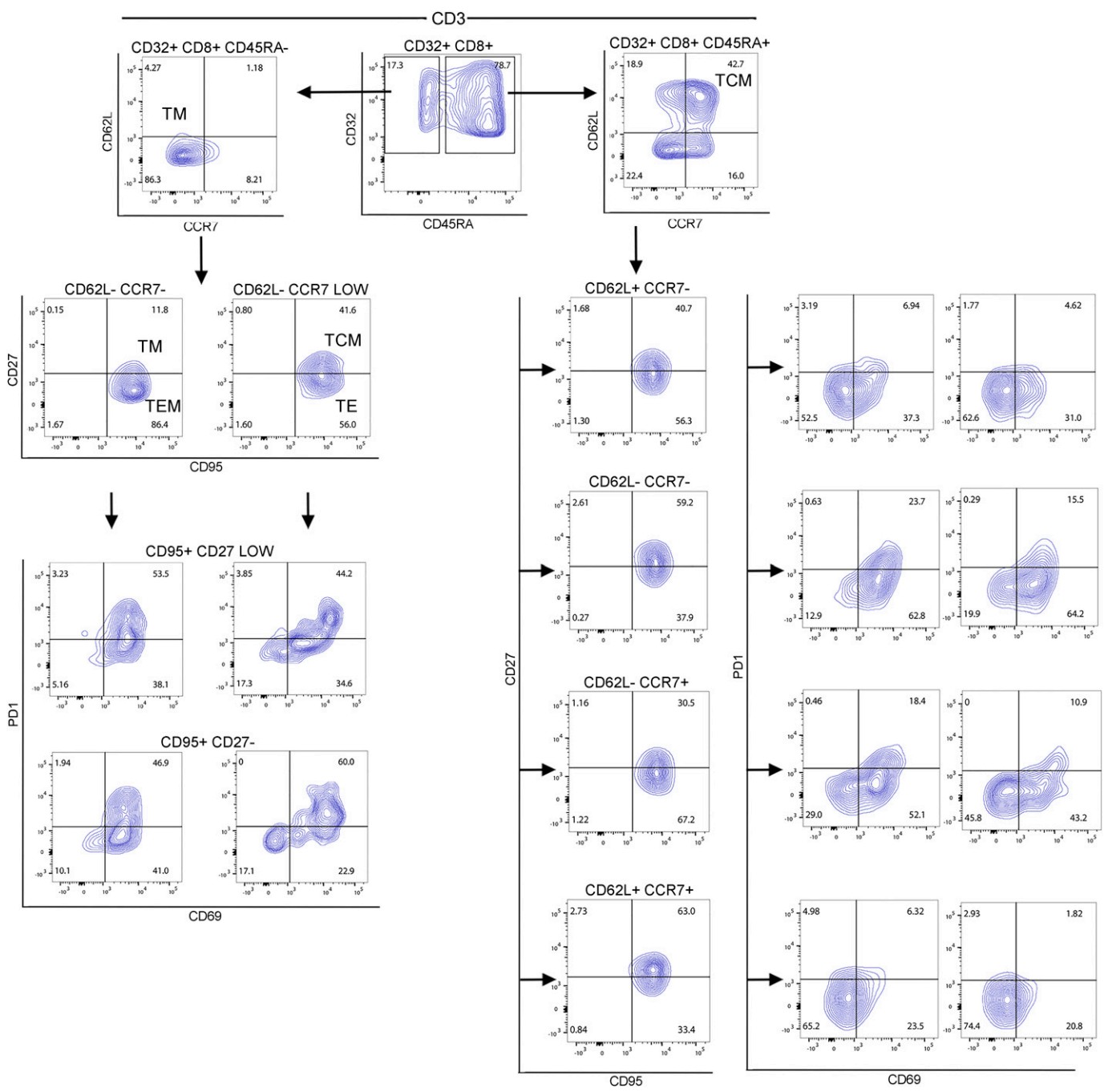

**Figure 2. Phenotypic analysis of CD32-CR CD8⁺ T cells.**
After a 3-wk activation with IL-7 and IL-15, CD32-CR CD8⁺ T cells were harvested and stained with 12 fluorescent mAbs specific for T-cell basic and differentiation markers. Then, cells were processed by a BD FACSLyric and the data were analyzed using FlowJo software. Data are representative of three experiments obtained with similar results.

while improving therapeutic mAb effects, CD32-CR also triggers antibody-independent T cell–mediated cytotoxicity.

We then assessed type and quality of cytokines released in the culture supernatants of CD32-CR T cells in the presence or absence of MDA-MB-468 cells. Fig 4B shows that supernatants of CD32-CR T cells contained proinflammatory and anti-inflammatory cytokines including RANTES, CCL-1, GM-CSF, and IL-13, respectively. In contrast,

NT T cells did not. Interestingly, except for RANTES, CD32-CR T cells in combination with MDA-MB-468 cells produced a significant enhancement of CCL-1, GM-CSF, IL-13, and IFNγ release, suggesting a direct involvement of the CD32-CR.

Recent studies have provided evidence that upon conjugation with leukemia or solid tumor cells, NK cells may undergo cellular aberrations including CD16A down-regulation and apoptosis

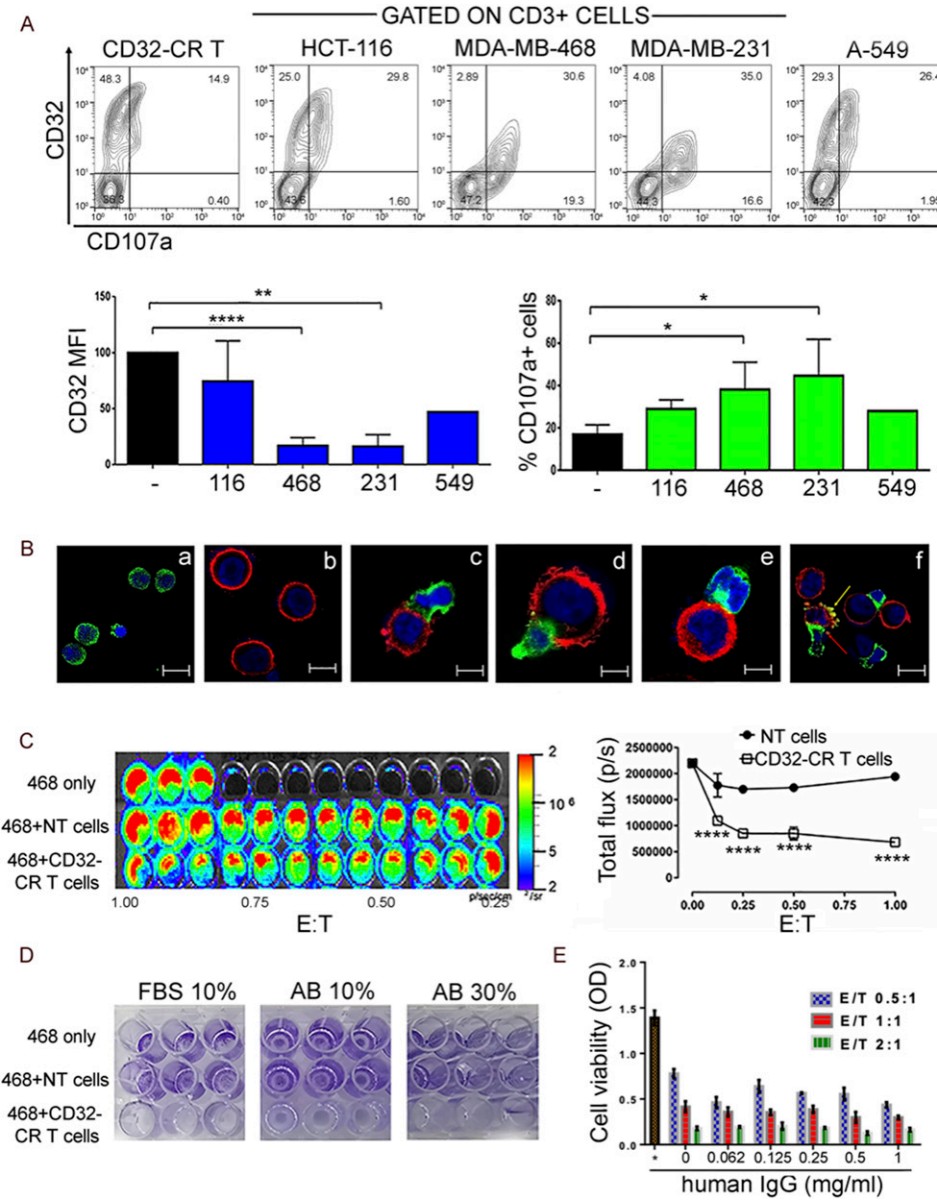

**Figure 3. CD32-CR T cells exert direct anti-tumor activity against triple-negative breast cancer cells.**
**(A)** Upper panel, CD32-CR T-cell down-regulation, and CD107a release (see the Materials and Methods section). CD32-CR T cells were stained as indicated and identified by gating on CD3⁺cells. Lower panel: cumulative analysis of CD32 down-regulation (left) and CD107a release (right) of three donors as induced by cancer cells. X-axes: abbreviations; MFI, mean fluorescence intensity. **(B)** Confocal imaging of CD32–CR T [(green)a] and MDA-MB-468 cells [(red)b] after a 2-h incubation (E:T 2:1) at 37°C shows the formation of lytic synapse consisting of firm adhesion (c), CD32-CR dipolar localization (d), internalization (e), and cell blebbing (arrows) (f). Images were acquired using Leica confocal microscope at magnification, 63X (oil immersion). **(C)** Image (left) and quantification (right) of light emission generated by luciferase-expressing cells through ATP-dependent conversion of luciferin to oxyluciferin used as a non-lytic, real-time cell viability assay. MDA-MB-468 cells were incubated for 72 h, at 37°C, with or without CD32-CR T cells at the indicated E:T ratios. NT T cells were used as a control. **(C, D)** MDA-MB-468 cells were incubated with engineered T cells or NT T cells at an E:T ratio of 2:1 as described in panel (C). Then, T cells and MDA-MB-468 non-adherent (dead) cells were removed, whereas adherent cells were stained with crystal violet. **(E)** Triplicates of MDA-MB-468 cells were incubated for 72 h at 37°C at different E:T ratios with or without scalar concentration of IgG as indicated. Then, non-adherent cells were removed and viable cells were assessed by the MTT assay as described in the Materials and Methods section. Data are representative of three experiments obtained with similar results. Values are expressed as mean ± SD. *$P < 0.05$; **$P < 0.01$; ****$P < 0.0001$.

leading to mutual effector-target cell elimination (Sconocchia et al, 2009, 2011; Arriga et al, 2016). In addition, our RNAseq and phenotypic studies suggest that, compared with NT T cells, CD32-CR T cells showed significantly higher pro-inflammatory and pro-apoptotic activation pathways. Indeed, following a 3-h co-incubation, MDA-MB-468–induced CD32-CR T down-regulation and CD32-CR T cell apoptosis (Fig 4C), suggesting that effector and target cells undergo mutual elimination in the absence of mAbs.

### CD32-CR T cell–mediated cytotoxicity is not limited to TNBC cells, does not involve primary myoblasts and fibroblasts, and protects CB17-SCID mice from the MDA-MB-468 cell challenge

The molecular classification of human BC relies on the expression of hormone receptors (HRs) and epidermal growth factor 2 (HER2)

on cancer cells. Hence, four molecular subtypes have been identified namely luminal A (HR⁺/HER2⁻), luminal B (HR⁺HER2⁺), HER2 enriched, and triple-negative (HR⁻/HER2⁻) (Reis-Filho & Pusztai, 2011). Based on this information and on previously shown data, we investigated whether the cytotoxic activity of the CD32-CR T cells was merely restricted to TNBC cell lines.

Therefore, engineered T cells were tested against four additional non-TNBC cell lines including HCC-1954 and SK-BR-3 (HER2 enriched) MCF-7 and T-47-D (luminal A) (Dai et al, 2017), and three primary myoblasts (h-MB), skin fibroblasts (BJ), and lung fibroblasts (IMR-90) human cell lines (Fig 4D).

The viability of the HCC-1954 cells was significantly affected by CD32-CR T cells (Fig 4D, upper right panel) although not as much as that of TNBC cells (Fig 4D, upper left and middle panels), whereas control NT T cells were fully inactive. It must be noted that the

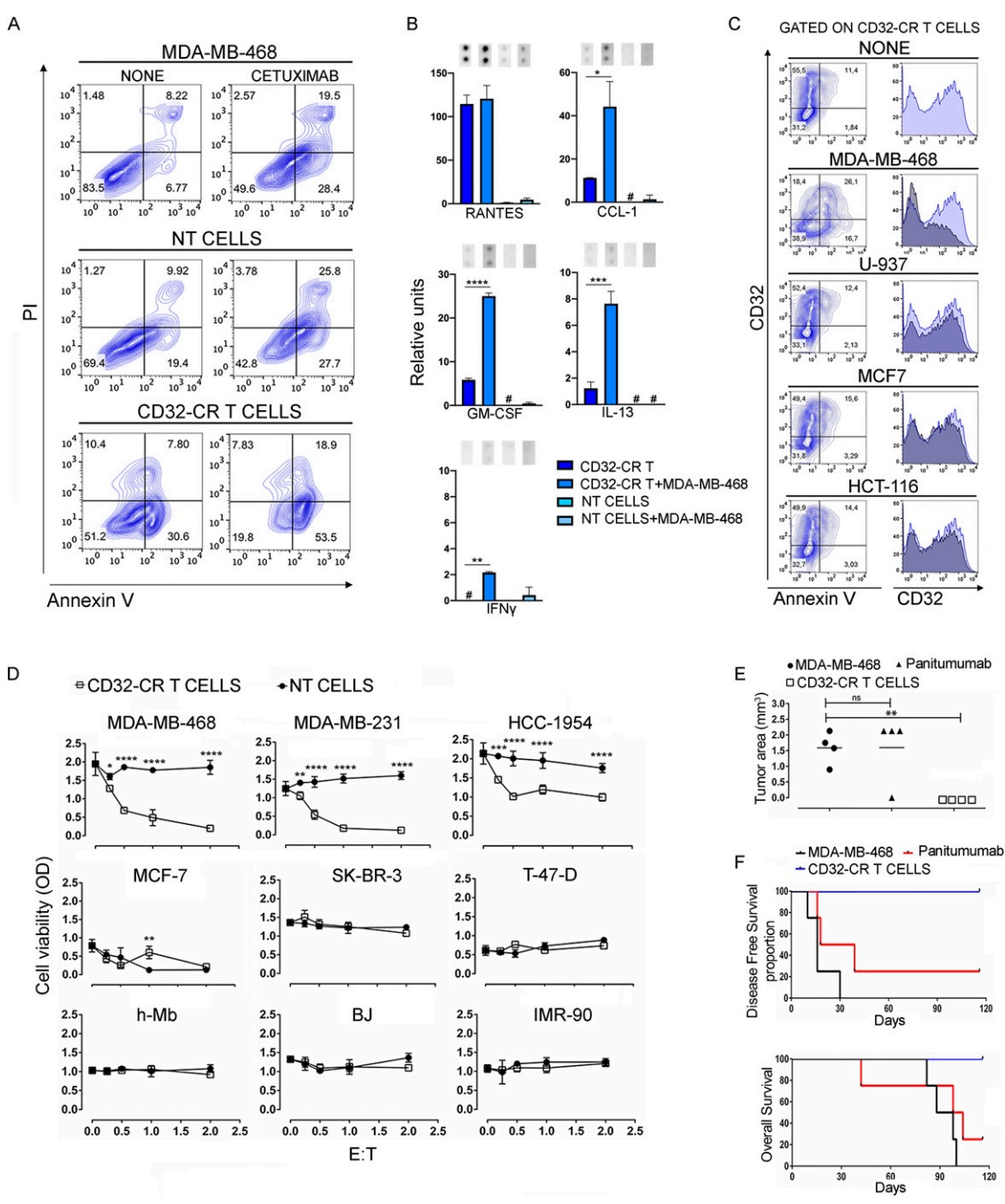

**Figure 4. CD32-CR T cell cytotoxicity and target cell specificity.**
**(A, C)** Following a 4-h incubation, CD32-CR T cells or NT T cells, in the presence or absence of the indicated cell lines, were harvested, stained with APC-anti-CD3 mAb, propidium iodide (PI), and/or FITC-annexin V, and analyzed by flow cytometry. **(A)** Contour plots showing the cytotoxic activity of CD32-CR T cells on MDA-MB-468 cells, in the presence or absence of cetuximab, were evaluated by gating on CD3-negative cells. Data are representative of five experiments independently performed. **(B)** Cytokine release by transduced or NT T cells, in the presence or absence of MDA-MB-468, was evaluated as described (see the Materials and Methods section). **(C)** Apoptosis induced by the indicated cancer cells in CD32-CR T cells was analyzed by gating on CD32-CR–positive cells. Data are representative of five experiments independently performed. **(D)** Preferential anti-cancer activity of CD32-CR T cells was not limited to TNB cells. BC cells or normal cells were incubated for 12 h, at 37°C in the presence or absence of CD32-CR T cells at the indicated E:T ratios. NT T cells were used as a control. Cell viability was evaluated by MTT assay. Values are expressed as mean ± SD. Anti-tumor activity of CD32-CR T cells (see the Materials and Methods section). Data are representative of three experiments independently performed. **(E)** Shows a scatterplot analysis of tumor volumes on day 120 post-injection. **(F)** Disease-free survival and overall survival analysis (see the Materials and Methods section). Data were analyzed by unpaired t test *$P < 0.05$, **$P < 0.01$, ***$P < 0.001$, ****$P < 0.0001$.

anti-tumor activity of CD32-CR T cells was impressive and even detectable at an E:T ratio lower than 0.5:1. In contrast, MCF-7, SK-BR-3, and T-47-D were resistant to CD32-CR T cells (Fig 4D, middle

panels). Most interestingly, neither engineered T cells nor NT T cells impacted the viability of the three primary, non-cancerous, cell lines (Fig 4D lower panels). These data suggest that the activity of

CD32-CR T cells is not restricted to TNBC cell lines, but it does not affect normal myoblasts and skin and lung fibroblasts.

To address the in vivo anti-tumor potential of CD32-CR T cells, three groups of CB17-SCID mice were injected subcutaneously with $1 \times 10^6$ MDA-MB-468 cells alone, with panitumumab (150 $\mu$g) or with $0.6 \times 10^6$ CD32-CR T cells. Tumor growth was monitored over time (Fig 4E). CD32-CR T cells inhibited the TNBC breast cancer cell growth in four of four mice ($P = 0.01$) whereas panitumumab had only a limited effect. As expected, whereas all untreated mice developed a measurable tumor within 30 d and had to be euthanized within 100 d, only 1 of four mice treated with panitumumab was disease-free and alive on day 120 of follow-up (Fig 4F, upper panel). In contrast, four of four mice treated with CD32-CR T cells were tumor-free and alive on day 120 (Fig 4F, lower panel). These data indicate that CD32-CR T cell–based immunotherapy is associated with a favorable course of the disease in the absence of therapeutic mAb administration.

### RNA-sequencing analysis identified transcriptomic signatures of 73 genes associated with susceptibility of breast cancer cells to CD32-CR T cell cytotoxicity

To gain insights into the molecular mechanisms underlying differential susceptibility of BC cells to CD32-CR T cells and to identify relevant transcriptional signatures, publicly available RNA sequences (RNA-Seq) were interrogated. RNA-Seq from MDA-MB-468, MDA-MB-231, and HCC-1954–sensitive and MCF-7, SK-BR-3, and T-47-D resistant cells was analyzed in detail. Of 60.469 genes tested, 1.418 (Table S2) were found to be differentially expressed in sensitive and resistant BC cell lines. Among these genes, 902 were up-regulated and 516 were down-regulated in sensitive versus resistant cells as summarized in the volcano plot (Fig 5A).

To identify key factor(s) conferring sensitivity to BC cells, we focused our attention on genes significantly up-regulated in the three sensitive BC cell lines. To further filter involved genes, this list was annotated based on the Gene Ontology Cellular component category, thus allowing the selection of 73 genes (Table S2) coding for cell surface proteins (Fig 5B). Then, the clinical significance of the overexpressed genes associated with the BC cell susceptibility to CD32-CR T cells was assessed by using a clinically well-documented TCGA advanced BC-RNA-seq database R2: Genomics Analysis and Visualization Platform (http://r2.amc.nl). We first analyzed the top 10 representative genes identified in the sensitive cells with the highest differential expression compared with that of resistant BC cells. Fig 5C shows that the expression level of these genes was significantly higher than that detected in the BC-resistant cells (bar graph panels). Importantly, a high expression of 7 of these 10 genes was significantly associated with improved survival in advanced BC patients. Only ROR 1 gene appeared to be associated with a poor prognosis. Subsequently, we extended this investigation to the other DEGs. Table 1 shows a direct association between a CD32-CR T cell susceptibility and additional 35 genes bringing the number of genes associated with favorable prognosis outcomes up to 42. Interestingly, these 42 genes were homogeneously expressed in 3 of 3 susceptible BC cells, whereas the remaining 31 genes were more heterogeneously expressed (Fig 5B). These data suggest that the sensitivity of BC cells to CD32-CR T cells

may predict a favorable prognosis in advanced BC prompting a previously unsuspected anti-tumor role in BC immunobiology. On the other hand, the prognostic significance of some of these genes has already been described while that of many other genes among them IL-7R and PCSK9 has not been reported yet.

### Role of the ICAM1 in the regulation of the susceptibility of BC cells to the CD32-CR T cell–mediated cytotoxicity

Among the 42 overexpressed genes identified, we were particularly interested in the role of ICAM1 because it is involved in a variety of immunological functions including T cell binding, immunological synapse formation, and activation (Bui et al, 2020).

To validate RNA-Seq study results, we assessed the distribution of ICAM1 protein expression among CD32-CR–sensitive and resistant cell lines by immunostaining with a PE-conjugated anti-CD54 mAb. Fig 6A shows that ICAM1 was preferentially expressed by MDA-MB-468, MDA-MB-231, and HCC-1954, whereas resistant cells were negative or expressed it to a lesser extent (MCF-7, SK-BR3, and T-47-D) than sensitive cells. The well-defined ICAM1 binding to LFA1 might suggest that the disruption of this interaction could protect tumor cells from the cytotoxic effects of CD32-CR T cells. Fig 6B shows that indeed, the addition to the cultures of an anti-CD18$\alpha$ mAb significantly reduced the susceptibility of BC cells to CD32-CR T cells in a dose-dependent manner (dose range 1.25–10 $\mu$g/ml). These results underline the key role of ICAM1/LFA1 interaction in the generation of CD32-CR T cell–mediated anti-tumor activity (Fig 7).

## Discussion

CD16A, CD32, and CD64 bind, with different affinities, to the Fc fragment of the Igs and pentraxins. The latter are a family of evolutionary conserved, soluble proteins present in the serum and mediate pro-inflammatory functions (Bottazzi et al, 2016). Interestingly, however, other than Fc and pentraxins, CD16 has been shown to bind putative surface ligand(s) on a variety of tumor cells leading to the activation of NK cells and cancer cell elimination (Mandelboim et al, 1999). However, likewise CD16, extracellular CD32 is composed of two immunoglobulin-like domains and binds with low affinity to the Fc fragment of Igs and pentraxins (Bharadwaj et al, 1999; Lu et al, 2008; Nimmerjahn & Ravetch, 2008). In the current state of knowledge, it is not known whether additional Fc$\gamma$Rs recognize cell surface ligand(s) on pathological and/or healthy cells in the absence of target-specific antibodies.

Here, we have used CD32-CR T cells to explore the localization of CD32 putative ligand(s) on the surface of 20 distinct cell types of which 15 were cancer cell lines. The molecular structure of CD32-CR makes it a cytotoxic triggering molecule when it is transduced in cytotoxic cells. Therefore, cross-linking of CD32-CR with its putative ligand(s), on the surface of the target cells, induces the activation of the T-cell killing machinery. Concordantly, we have demonstrated that CD32-CR T cells recognize CD32-CR putative ligand(s), on MDA-MB-468 and MDA-MB-231 BC cells, resulting in specific CD32 down-regulation and CD107a release consistent with cytotoxic granule

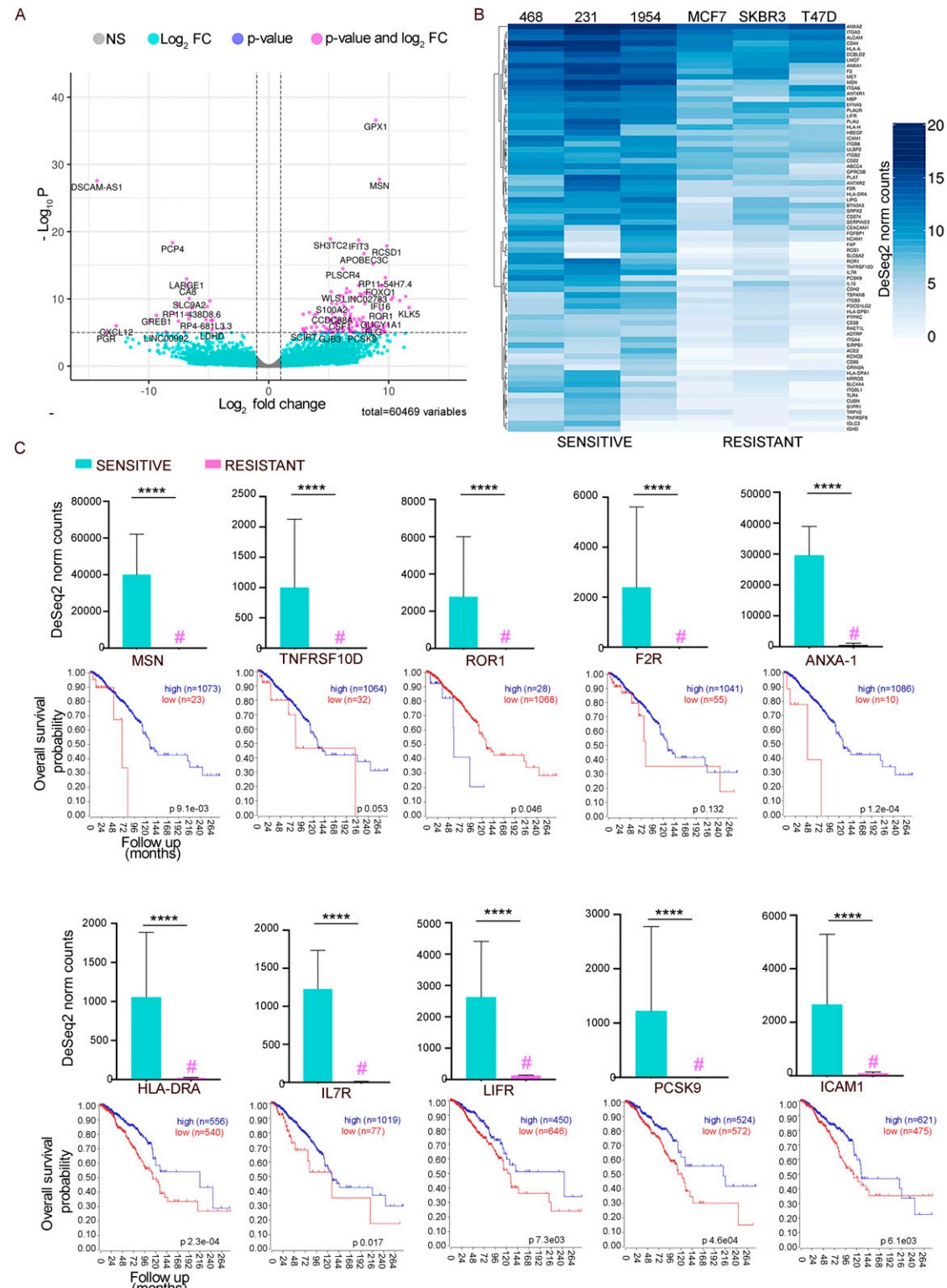

**Figure 5. Sensitivity to CD32-CR T cells cytotoxicity is associated with up-regulated expression of genes correlating with favorable prognosis in advanced BC patients.**
**(A)** Volcano plot showing differential transcriptome of BC cells with different sensitivity to CD32-CR T cell–mediated cytotoxicity analyzed in terms of significance levels and fold changes in gene expression, as estimated from DESeq2 analysis of RNA-Seq data. A total of 60,469 genes were analyzed and 1,418 were found differentially expressed in sensitive and resistant BC cell lines. Among these genes, 902 were up-regulated and 516 down-regulated in sensitive versus resistant BC cells. The magenta points represent genes with $\log_2$FC > |2| and $P$.adj < 10 × 10$^{-6}$. **(B)** Heat map showing the expression levels of 73 genes encoding cell surface proteins, selected based on the

exocytosis of T cells in an HLA-independent manner (Alter et al, 2004). The direct anti-tumor activity is not restricted to the triple-negative BC cell lines but also involves HER2+ HCC-1954 cells, CRC HT29, and glioblastoma U-373MG (data not shown), whereas HUVEC cells induced CD32-CR down-regulation. In contrast, no significant evidence of CD32 down-regulation is detectable upon conjugation with hematopoietic cells, a variety of epithelial cancer cells, two normal, primary fibroblasts, and one myoblast cell line, indicating that CD32 ligand(s) is preferentially expressed on BC cells.

The anti-tumor properties of CD32-CR T cells have been confirmed by more than four sets of in vitro assays including confocal microscopy, bioluminescence, flow cytometry, and MTT assays. The observation that three of seven BC cell lines were sensitive to the engineered T cells suggests the existence of a subgroup of BC cells resistant to CD32-CR T cell–mediated cytotoxicity.

The use of biosensors for the identification and localization of surface molecules with unknown ligand(s) or function is a reliable methodology and is consistent with our previous work in which bispecific monoclonal antibodies (BsAbs) allowed the identification of CD44 as a cytotoxic triggering molecule on NK cell and its metabolic pathways (Sconocchia et al, 1994, 1997; Pericle et al, 1996).

One important issue is whether the direct cytotoxic activity of engineered T cells has any relationship with ADCC. We are comfortably convinced that CD32-CR–mediated cytotoxicity is an independent function because either human serum or purified IgGs failed to protect sensitive EGFR-positive BC cells from CD32-CR T cells, but rather their killing was enhanced in the presence of cetuximab. The results shown in the present study indicate that CD32 putative ligand(s) and Igs detain distinct docking sites on extracellular CD32 and support the role of CD32-CR–mediated cytotoxicity independent from ADCC. Also, BC cell killing has been achieved at a very low level of E:T ratio such as 0.5:1 even when the CD32-CR T cell transduction has been lower than 100% indicating that the extent of CD32-CR T cell anti-tumor activity is strikingly powerful.

Notably, we have observed that specific conjugation of CD32-CR T cells with sensitive BC cells led to mutual cell killing, whereas NT T cells did not. These results are consistent with published results indicating epithelial cancer cells and leukemia cells induced NK cell killing and CD16A down-regulation (Sconocchia et al, 2009, 2012; Arriga et al, 2016). Although the mechanism underlying this phenomenon is not readily apparent, it might be possible to hypothesize a feedback mechanism aimed to turn off an overwhelming immune response mediated by cytotoxic cells.

Although the anti-cancer activity of CD32-CR T cells is preferentially directed towards BC cells, we have also evidence that some CRC and glioblastoma cell lines can be targeted (data not shown), whereas most hematopoietic malignant cells, we have used, are resistant (Marei et al, 2019). Based on the susceptibility of cancer cells to CD32-CR T cells, we have identified two populations of BC cells defined as sensitive or resistant. The RNA-seq analysis of the BC cell lines has underlined the compelling difference in gene expression between sensitive and resistant BC representing a remarkable transcriptomic signature for the identification of target cells. Among 42 DEGs, ICAM1 has come to our attention. ICAM1 is a key molecule involved in the leukocyte transmigration through the endothelial barrier but is also noteworthy that ICAM1 is heterogeneously expressed in BC tissues (Rosette et al, 2005). Therefore, the reduced susceptibility of sensitive BC cells to CD32-CR T cell activity by the anti-CD18 mAb is likely to be due at least in part to the blockade of the interaction of β2 integrins on T cells (Bui et al, 2020) with ICAM1 leading to the inhibition of CD32-CR T-cell cytotoxicity.

Although ICAM1 appears to be required for the elicitation of an efficient CD32-CR T cell anti-cancer activity (Fig 6) its expression is non-sufficient for the following considerations. First, optimal CD32-CR direct cytotoxicity requires the presence of significant expression of ICAM1 and CD32-CR ligand(s) as indicated by CD32-CR profound down-regulation in sensitive BC cells. Second, ICAM1-positive U937 and ML-2 cells have proven to be resistant to CD32-CR T cells, suggesting they lack CD32-ligand(s). Third, the absence of ICAM1 on resistant BC cells does not exclude the presence of CD32-CR ligand(s) on these cells. In addition, the finding that ICAM1-positive HUVEC cells induced CD32-CR down-regulation is evidence of the presence of CD32-ligand(s) on endothelial cells. In contrast, no significant CD32-CR down-regulation has been observed following cell-to-cell contact of CD32-CR T cells with ICAM1-negative cells, including HCT116 and A549.

The information provided by this study raises some basic questions. What is the identity of CD32-CR ligand (s)? Is there any cross-talk between normal epithelial/endothelial cells and CD32-positive innate cells and what type of cellular response(s) is (are) generated? We are motivated to keep working on these issues because the availability of this information may have a relevant impact in the field of the biology of innate cells.

Nevertheless, despite the incomplete comprehension of the nature and distribution of CD32 putative ligand(s), CD32-CR T cell–based immunotherapy, alone or in association with anti-EGFR therapeutic mAbs (Caratelli et al, 2020), might be used for designing rational immunotherapies for targeting BC as suggested by in vivo results.

Safety concerns are raised by the fact that HUVEC cells have induced significant down-regulation of CD32-CR. One may predict that engineered T cell immunotherapy may lead to endothelial cell damage reminding the capillary leak syndrome described during LAK cell–based and cluster differentiation mAb immunotherapy (Jeong et al, 2019) and in 19-CAR T cell immunotherapy. However, similarly to classic CAR T cells, the safety of CD32-CR T cells might be enhanced by using suicide genes. Unfortunately, an optimal animal model capable of providing reliable information about the safety of this type of immunotherapies is not available yet. It would be useful to determine whether the behavior of mouse and human FcγCR coincide allowing a more detailed investigation of CD32-CR T cell off-shelf toxicity in syngeneic tumor-bearing mice models.

In addition, the identification of a gene signature associated with BC cell sensitivity to CD32-CR T cells and predicting the favorable

Gene Ontology Cellular component category significantly up-regulated in sensitive versus resistant BC cells. **(C)** The 10 genes with the most significant *P*.adj were selected and their expression values were plotted as DESeq2-normalized counts (mean ± standard error). For each gene, a Kaplan–Meier curve, analyzing their prognostic significance as observed in 1.097 breast invasive carcinomas from TCGA, is shown (https://hgserver1.amc.nl/cgi-bin/r2/main.cgi). The expression levels of 7 of 10 genes are significantly associated with a favorable prognosis in patients with breast invasive carcinomas. ****P*.adj ≤ 0.0001 (see the Materials and Methods section).

**Table 1.** Breast cancer cell sensitivity to CD32-CR T cell–mediated cytotoxicity identifies overexpression of 42 genes associated with enhanced overall survival in advanced breast cancer patients.

| Gene name | log$_2$FoldChange expression of sensitive versus resistant | Expression *Padj* | Survival *P*-value |
|---|---|---|---|
| MSN | 9.234237685 | $1.61 \times 10^{-28}$ | 0.0091 |
| ANXA1 | 5.680103532 | $3.59 \times 10^{-7}$ | 0.00012 |
| HLA-DRA | 5.553333035 | $1.54 \times 10^{-6}$ | 0.00023 |
| IL-7R | 7.847016223 | $2.66 \times 10^{-6}$ | 0.017 |
| LIFR | 4.314342941 | $7.18 \times 10^{-6}$ | 0.0073 |
| PCSK9 | 7.797806132 | $9.48 \times 10^{-6}$ | 0.00046 |
| ICAM1 | 4.71910841 | $9.80 \times 10^{-6}$ | 0.0061 |
| SLC4A4 | 7.233043078 | $1.02 \times 10^{-5}$ | 0.031 |
| CD44 | 5.135518877 | $1.05 \times 10^{-5}$ | 0.000000034 |
| MBP | 5.863942435 | $1.05 \times 10^{-5}$ | 0.0091 |
| PDCD1LG2 | 5.564103601 | 0.000134633 | 0.0008 |
| CD38 | 3.655160914 | 0.000253909 | 0.0094 |
| ANTXR1 | 4.670855734 | 0.000265847 | 0.00014 |
| CUBN | 6.381850551 | 0.000307575 | 0.002 |
| PLAT | 6.696990062 | 0.000432311 | 0.00078 |
| FGFBP1 | 7.031398988 | 0.000463226 | 0.001 |
| HLA-DPB1 | 3.899732307 | 0.000538005 | 0.0028 |
| ITGB8 | 4.614246358 | 0.000589962 | 0.014 |
| LIPG | 5.903021878 | 0.001436942 | 0.00031 |
| HLA-A | 4.313972391 | 0.001450444 | 0.016 |
| HLA-DPA1 | 4.763693695 | 0.001638118 | 0.006 |
| SRPX2 | 4.126912066 | 0.002044177 | 0.00041 |
| NRROS | 5.079683458 | 0.005607296 | 0.032 |
| CDH2 | 3.75195309 | 0.005748377 | 0.018 |
| HBEGF | 4.941781375 | 0.00582612 | 0.012 |
| SLC6A2 | 8.393035966 | 0.006385352 | 0.00081 |
| BTN3A3 | 3.491327553 | 0.00694209 | 0.002 |
| CEACAM1 | 5.103805021 | 0.007057192 | 0.000002 |
| ITGA3 | 3.329192706 | 0.007409475 | 0.0069 |
| ITGB2 | 2.909231769 | 0.0095556 | 0.011 |
| SERPINE2 | 3.589186779 | 0.010257949 | 0.024 |
| ADTRP | 3.829816509 | 0.01390029 | 0.0038 |
| FAP | 5.764241237 | 0.021549094 | 0.00033 |
| DCBLD2 | 3.035651524 | 0.024155054 | 0.02 |
| PTPRC | 3.018702502 | 0.024542382 | 0.013 |
| CD22 | 3.04395917 | 0.024835513 | 0.00088 |
| HLA-H | 4.521019966 | 0.026629949 | 0.0045 |
| TNFRSF9 | 4.538715226 | 0.029923057 | 0.011 |
| MET | 3.95185859 | 0.030399118 | 0.0014 |
| IL-15 | 3.471780313 | 0.036004979 | 0.024 |
| CD274 | 4.168703408 | 0.037513963 | 0.0037 |
| TRPV2 | 4.070284638 | 0.038141506 | 0.012 |

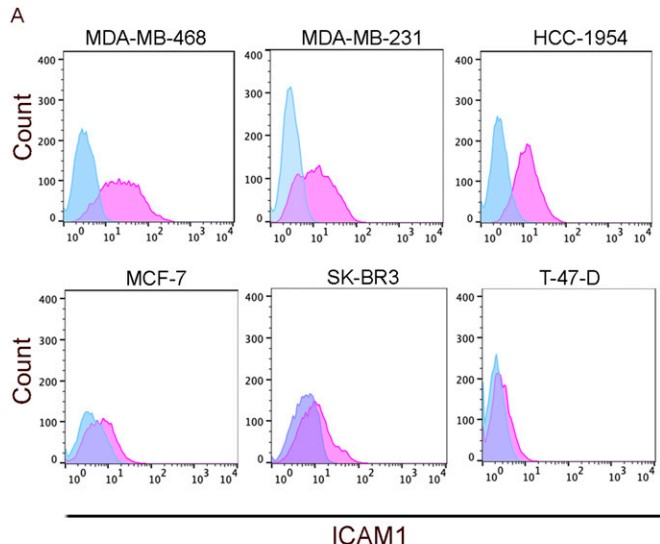

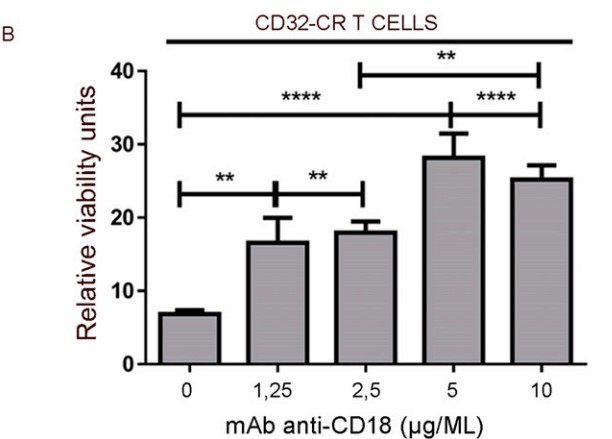

**Figure 6.  ICAM1 is preferentially expressed on the surface of sensitive BC cells and regulates CD32-CR T-cell anti-cancer activity.**
**(A)** Expression of ICAM1 on the surface of sensitive (upper panel) and resistant (lower panel) was assessed by immunostaining using a PE-conjugated anti-human ICAM1 (CD54). After a 30-min incubation, cells were washed and analyzed by flow cytometry. **(B)** Rescue of MDA-MB-468 cell viability by LFA1-ICAM1 blockade was tested as follows: CD32-CR T cells were pre-incubated with scalar concentrations of anti-CD18 mAb for 30 min ant 37°C and washed. Then FcR negative MDA-MB-468 were added at an E:T ratio of 1:1. After 2 d, non-adherent cells were removed and the viability of MDA-MB-468 was measured by the MTT assay. Data are representative of three experiments independently performed. **$P < 0.01, ****P < 0.0001.

clinical course of BC in a retrospective study of 1,400 patients with advanced disease opens new unsuspected investigation pathways in the field of innate cell and tumor biology.

# Materials and Methods

### Antibodies and reagents

Allophycocyanin (APC)-conjugated mouse anti-human CD3 and anti-human CCR7, APC-Cy7–conjugated mouse anti-human CD69, APC-R700–conjugated mouse anti-human LAG-3, fluorescein iso-thiocyanate (FITC)–conjugated mouse anti-human CD3, anti-CD62L, and anti-CD107a, phycoerythrin (PE)–conjugated mouse anti-human CD32 and anti-CD54, PE-Cyanine (PE-Cy)7 mouse anti-human PD1, PerCP-Cy5.5–conjugated mouse anti-human CD45RA, BDHV450-conjugated mouse anti-human CD95, BDHV500 anti-human CD27, BDHBV605 anti-human CD4, BDHBV711 anti-human CD45RO, BDHBV786 anti-human CD8, purified NA/LE mouse anti-human CD3, purified NA/LE mouse anti-human CD28, purified mouse anti-human CD32 (8.26), and purified mouse anti-human CD18 were purchased from BD Bioscience. Rabbit anti-human EGFR (D3B1) was purchased from Cell Signaling Technology. Cy-5–conjugated donkey anti-rabbit IgG and Alexa Fluor-488 F(ab')2 fragment goat anti-mouse IgG were purchased from Jackson ImmunoResearch Laboratories and Invitrogen, respectively. Rabbit anti-asialo-GM1 was purchased from Wako Chemical Europe Monensin (M5273), 3-(4,5-dimethylthiazol-2-yl)-2,5-diphenyltetrazolium bromide (MTT), Hoechst nuclei staining, fluoromount aqueous mounting medium, and DMSO were purchased from Sigma-Aldrich and GeneJuice Transfection Reagent (Novagen) from Millipore. Human recombinant IL-7 and IL-15 were purchased from Miltenyi Biotec. Retronectin (Recombinant Human Fibronectin) was purchased from Takara Bio. D-luciferin was purchased from PerkinElmer. DMEM, IMDM, EMEM, McCoy's 5A, and RPMI 1640 media were purchased from Lonza. FBS, human AB serum, Phosphate-buffered solution (PBS), L-glutamine, and penicillin/streptomycin were purchased from Euroclone. Complete media (CM) were supplemented with 10% FBS, 2 mM L-glutamine, 0.1 mg/ml streptomycin, and 100 U/ml penicillin.

### Cell lines

The packaging cell line 293T was cultured in IMDM CM and was used to generate helper-free retroviruses for T-cell transduction. Colo-rectal cancer (CRC) cell lines HCT116 and HT-29 were maintained in RPMI CM. Non-small-cell lung cancer (NSCLC) cell line A549, TNBC cells MDA-MB-231 and MDA-MB-468, SUM159 luminal A BC cells, MCF-7 and T-47-D (ER⁺, PR⁺/⁻, and HER2⁻), and HER2-enriched SKBR-3 and HCC-1954 (ER⁻, PR⁻, and HER2) were kindly provided by Dr. Antonio Rossi and Dr. Maria Lucibello (Institute of Translational Pharmacology, CNR). NSCLC cell line A549 and all BC cell lines were mantained in DMEM CM. Submandibular gland squamous cell carcinoma cell line A-253, hypopharyngeal squamous cell carcinoma cell line FaDu, acute myeloid leukemia cells ML-2, histiocytic lymphoma cells U937, lymphoblastic leukemia cells Jurkat, and human umbilical vein endothelial cell (HUVEC) were obtained from our laboratory cell collection, and maintained in McCoy's 5A, EMEM, RPMI CM, respectively. Human fibroblasts IMR-90 and BJ, and human myoblasts were kindly provided by Dr. Lucia Latella (Institute of Translational Pharmacology, CNR) Table 1. All the cell lines were mycoplasma-free. Human myoblasts were grown in DMEM supplemented with Primary Skeletal Muscle Growth Kit (ATCC). MDA-MB-468-Luc⁺ cells were transduced at a MOI 10 with a third-generation self-inactivating lentiviral vector expressing firefly luciferase as previously described (Di Rocco et al, 2012). Quantification of light emission generated by luciferase-expressing cells through ATP-dependent conversion of luciferin to oxyluciferin can be used as a nonlytic, real-time cell viability assay.

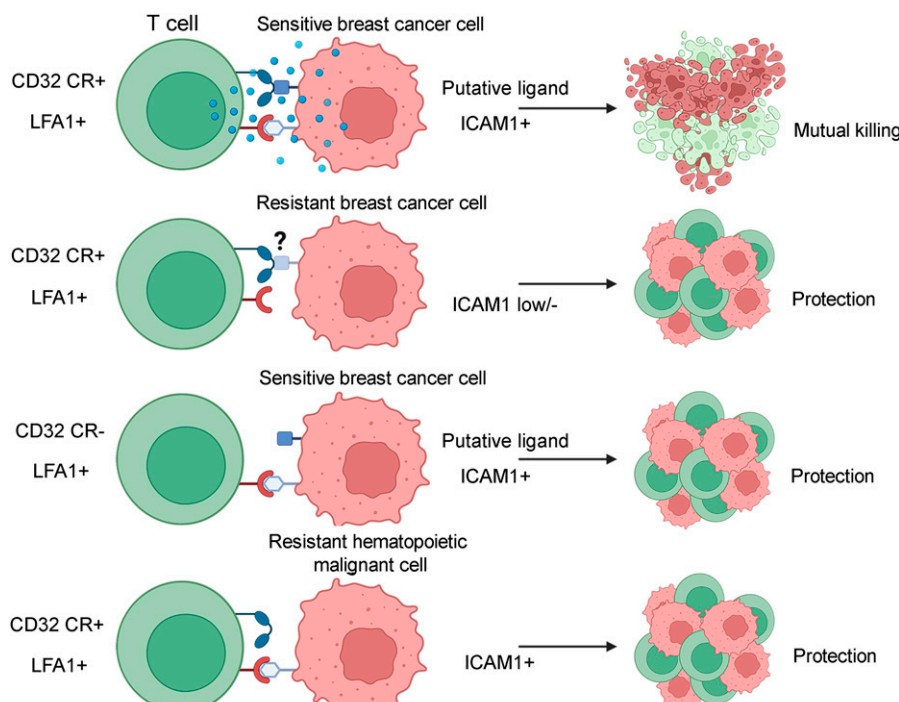

**Figure 7. Hypothetical representation of cellular events occurring following conjugation between CD32-CR T cells with sensitive or resistant cancer cells.**
Optimal activation of CD32-CR requires the expression of CD32 putative ligand(s) and ICAM1 on sensitive BC cells. Resistant BC cells are not susceptible to CD32-CR T cells because they may express the ligand(s) but lack ICAM1 or both the ligand(s) and ICAM1. CD32-CR–negative NT T cells do not kill CD32-CR–sensitive BC cells, despite their expression of ICAM1 and putative CD32 ligand(s). Also, CD32-CR–positive T cells do not kill ICAM1⁺ hematopoietic malignant cells because of defective expression of CD32 CR ligand(s).

## CD32-chimeric receptor production and T-cell transduction

The construction of the CD32-CR, virus production, and T cell transduction have been previously described (Caratelli et al, 2020). Briefly, HEK 293T cells were cotransfected with CD32-CR-SFG, PegPam, and RDF vectors. After 48 and 72 h, cell supernatants containing viral particles were harvested and stored at −80°C. Human PBMCs were isolated from buffy coats and seeded in 24-well plates precoated with anti-CD3 and anti-CD28 mAbs for 48 h. Activated T cells were collected and incubated with retroviral particles carrying CD32-CR for 72 h into a retronectin-coated 24 well-plate. After transduction, T cells were kept in culture for 12–15 d before using for further experiments. Transduced and non-transduced T cells were maintained in RPMI CM supplemented with 5 ng/ml IL-15 and 10 ng/ml IL-7.

## Single-cell RNA-seq analysis

Data pre-processing was performed in Cell Ranger, the default analysis pipeline recommended by 10x Genomics, to de-multiplex raw base call files to FASTQ files, and subsequently align reads to the human genome (GRCh 38) with the default parameters. The NT T and CD32-CR T samples were merged into a single matrix, and data were loaded into Seurat V4 for analysis (Hao et al, 2021). Cells with fewer than 200 detected genes, as well as cells with mitochondrial genes proportion higher than 10%, were excluded. After log-normalization, confounding factors including several detected genes and mitochondrial gene proportion were regressed.

To explore the transcriptomics profiles, we performed the clustering and dimension reduction analysis by pooling all cells together. Especially, the top 2,000 highly variable genes were obtained, and principal component analysis was performed. The top 20 Principal components (PCs) were used to perform clustering (with a resolution of 0.5) based on the k-nearest neighbors method in Seurat V4. The non-linear dimensional reduction technology uniform manifold approximation and projection (UMAP) was used to place study cells together in the two-dimensional space (based on 20 PCs).

Sample-specific up-regulated genes were determined by "FindAllMarkers" function in Seurat (only.pos = TRUE, logfc.threshold = 0.25; $P$_val_adj = $10^{-5}$). We generated heat maps displaying the expression profiles of up-regulated genes for each sample using "ComplexHeatmap" R package (Gu et al, 2016). To identify an enriched pathway for each sample, we identified sample-specific genes and over-expressed genes as defined above and performed pathway enrichment using the enricher tool (Kuleshov et al, 2016) using the HALLMARK option with test type being set as Fisher's exact test. Significant pathways were identified at an adjusted $P$-value < 0.01. The proapoptotic, apoptotic, and inflammatory pathways levels for each cell were calculated using the *AddModuleScore* function in Seurat. The proapoptotic pathway score was based on genes including *BAX*, *BAK1*, *BCL2L1*, *BAD*, *BID*, *BIK*, and *HRK*. The apoptotic pathway score was based on genes including *BCL2*, *BCL2L1*, *BCL2L2*, *BFLM*, *IQSEC2*, *MCL1*, and *BCL2A1*. The inflammatory pathway score was based on the gene set downloaded from https://www.gsea-msigdb.org/gsea/msigdb/cards/HALLMARK_INFLAMMATORY_RESPONSE.

## RNA-seq analysis of breast cancer cell lines

RNA-Seq from six breast cancer cell lines was downloaded from the NCBI SRA portal as part of the Cancer Cell Line Encyclopedia (SRP186687) project and transformed into fastq files. All the

preprocessing steps to trim adapters and remove low-quality reads were performed with the dedicated fastp (Chen et al, 2018) tool version 0.21.0. GRCh37/hg19 human reference genome was used for all analyses. Fastq files were aligned to the reference genome using HISAT2 (Kim et al, 2019) version 2.2.1 and parameters –no-mixed –no-discordant –very-sensitive –known-splicesite-infile –dta-cufflinks -x genome_snp_tran. Alignments in SAM format were converted into the binary BAM format, sorted by genomic coordinates, and indexed by SAMtools. Gene expression quantification was obtained with the featureCounts tool of the Subread (Liao et al, 2014) 2.0.2 package (parameters "--countReadPairs" and "-p") and the gencode annotation v38lift37. Counts' normalization and differential gene expression analysis was performed using the well-known Bioconductor package in R DESeq2 (Love et al, 2014) version 1.32.0 following the manufacturer's instructions. Gene set enrichment analysis was performed using the "Investigate Gene Set" tool from the GSEA web portal (https://www.gsea-msigdb.org) and the GO Cellular Component ontology. Plotting and statistics were performed using specific R packages.

### Confocal microscopy

CD32-CR T cells were cocultured with MDA-MB-468 breast cancer cells at an effector: target (E:T) ratio of 2:1 on poly-d-lysine (0.02%) pre-coated glass multi-chambers well. After a 2-h incubation, cells were fixed with a solution containing 4% of PFA. Samples were then washed and incubated with an FcR blocking reagent to avoid nonspecific staining. Cells were incubated with rabbit anti-human EGFR and mouse anti-human CD32 antibodies (1:50 dilution) for 1 h at room temperature in a humidity chamber. Cells were then stained with cy-5-conjugated donkey anti-rabbit and Alexa Fluor-488 goat anti-mouse secondary antibodies, whereas cell nuclei were counterstained with Hoechst for 5 min at RT. Samples were acquired using a 63X oil immersion lens on a Leica Laser Scanning confocal microscope (Leica Microsystems).

### Cytokine array

Cytokine array analysis has been previously described (Sconocchia et al, 2004, 2014a). CD32-CR T cells and NT T cells ($2 \times 10^5$) were cultured with or without MDA-MB-468 cells at an E:T of 2:1 in the absence of IL-7 and IL-15. After 72 h of incubation, culture supernatants were collected and the levels of cytokine release were assessed by using a duplicate 42 human cytokine antibody array (Ray Biotech).

### In vitro tumor cell viability assay

Antitumor activity of CD32-CR T cells was evaluated in vitro by a cell viability assay using an MTT reagent. Tumor target cells ($20 \times 10^3$/well) were seeded in triplicate in 96-well plates, and CD32-CR or non-transduced T cells were added at different E:T ratios. After a 12-h incubation at 37°C, non-adherent T cells were removed and the remaining adherent target cells were incubated with 100 $\mu$l of fresh medium supplemented with 20 $\mu$l of MTT (5 mg/ml) for 3 h at 37°C. Supernatants were then removed and 100 $\mu$l of DMSO were added

to each well and placed on a plate shaker for 15 min protected from light. Absorbance was measured at 570 nm.

For in vitro Bioluminescent Imaging (BLI), MDA-MB-468 luciferase-expressing cells were seeded in 96-well microplates in triplicate in presence of CD32-CR or NT T cells for 72 h at 37°C at different E:T ratios. The cell culture medium was then supplemented with D-luciferin (PerkinElmer) dissolved in PBS (150 $\mu$g/ml) for 10 min and analysis was performed using the IVIS Lumina II platform (PerkinElmer). Photons emitted from luciferase-expressing cells in selected regions of interest were quantified using the Living Image software.

### CD107a assay

CD107a release was tested by incubating CD32–CR T cells with A549, HCT116, MDA-MB-468, and MDA-MB-231 target cells at an E:T of 2:1 for 1 h at 37°C in a 5% $CO_2$. FITC mouse anti-human CD107a antibody diluted 1:50 was added in a final volume of 200 $\mu$l in a 96 multi-well plate. Then, 2 $\mu$M of monensin (Golgi stop) was added to each well and the cell co-culture was incubated for 4 h at 37°C. After incubation, samples were stained with PE mouse anti-human CD32 for 30 min at 4°C, washed, and fixed in 1% PFA. The CD107a expression on CD32-CR T cell surface was assessed by using a 2-laser BD FACSCalibur flow cytometer (Becton Dickinson). Results were analyzed using Tree Star, Inc. FlowJo software.

### Xenograft mouse model

In vivo experiments were performed following European Directive 2010/63/EU guidelines and regulations. The Italian Ministry of Health approved animal handling and procedures (authorization code: 186/2016-PR). Antitumor activity of CD32-CR T cells was assessed using 8-wk-old male CB17-SCID mice (CB17/lcr-PrkdcSCID/lcrlcoCrl, Charles River Laboratories, Cat. no. CRL:236, RRID: IMSR CRL:236), 12–18 g body weight, engrafted with MDA-MB-468 TNBC cells. Mice were housed in temperature-controlled rooms with a 12 h light/dark cycle and free access to sterile water and autoclaved standard chow diet (4RF25; Mucedola). Endogenous NK cell activity was suppressed by intraperitoneal injection of a 20 $\mu$l rabbit anti-asialo-GM1 antibody. Mice received anti-asialo-GM1 antibody on days −3, 0, +14, and +21 since tumor cell engraftment. Three groups of CB17-SCID mice (N = 4 per group) received 1.0 $\times 10^6$ MDA-MB-468 cells subcutaneously, in the right flank, with or without panitumumab (150 $\mu$g) or 0.6 $\times 10^6$ CD32-CR T cells at an E:T of 0.6:1. Every 3 d, tumor volumes (TV) were measured with caliper and calculated using the formula: TV ($cm^3$) = $4/3pr^3$, where r = (length + width)/4. Mice were euthanized when tumor volume reached 2 $cm^3$.

### Statistical analysis

The unpaired $t$ test or the Mann–Whitney test was used to analyze the results. Survival time differences were determined using the Kaplan–Meyer method and the log-rank (Mantel–Cox) test. Differences with a $P$-value < 0.05 were considered statistically significant.

## Data Availability

All data, code, and materials used in the analysis are available to any researcher for purposes of reproducing or extending the analysis. Some material transfers might be regulated by materials transfer agreements. Most of the data are available in the main text or supplementary material.

## Supplementary Information

## Acknowledgements

We thank Dr. Mauro Cozzolino and Aymone Gurtner for confocal microscopy technical assistance and Dr. Antonio Rossi for the helpful discussion. We also thank Dr. Pamela Papa and Dr. Matilde Paggiolu for administrative assistance. Italian Association for Cancer Research (AIRC) Foundation: Investigator Grants 2015-17120, Italian Association for Cancer Research (AIRC) Foundation: Investigator Grants 2020-24440, and European Union, European Fund for Regional Development, MUR-PON: Unit IFT, Grant TITAN 2021-ARS01_00906 to G Sconocchia.

### Author Contributions

G Sconocchia: conceptualization, data curation, funding acquisition, validation, investigation, visualization, methodology, and writing—original draft, review, and editing.
G Lanzilli: data curation, investigation, and visualization.
V Cesarini: data curation, visualization, methodology, and writing—original draft.
DA Silvestris: investigation, visualization, and methodology.
K Rezvani: investigation, visualization, and methodology.
R Arriga: data curation and investigation.
S Caratelli: investigation and visualization.
K Chen: investigation and methodology.
J Dou: investigation and visualization.
C Cenciarelli: data curation, investigation, and visualization.
G Toietta: data curation, investigation, visualization, and methodology.
S Baldari: data curation, investigation, visualization, and methodology.
T Sconocchia: investigation, visualization, and writing—original draft.
F De Paolis: investigation and visualization.
A Aureli: writing—review and editing.
G Iezzi: writing—review and editing.
M Irno Consalvo: investigation and visualization.
F Buccisano: investigation and visualization.
MI del Principe: writing—review and editing.
L Maurillo: writing—review and editing.
A Venditti: writing—review and editing.
A Ottaviani: data curation, investigation, visualization, and writing—original draft.
GC Spagnoli: writing—original draft.

### Conflict of Interest Statement

CNR/La Sapienza, Patent no. 102014902258369 (Ex TO2014A000361) filed on May 6, 2014, Innate Chimeric Receptors for Immunotherapy awarded on July 12, 2019 to G Sconocchia and S Caratelli. K Rezvani and The University of Texas MD Anderson Cancer Center have an institutional financial conflict of interest with Takeda Pharmaceutical and Affimed GmbH. K Rezvani participates on the Scientific Advisory Board for GemoAb, Avenge Bio, Virogin Biotech, GSK, Caribou Biosciences, Navan Technologies, and Bayer.

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
