## [Reviewer comments · Life Science Alliance]

Direct CD32 T cell cytotoxicity: implications for breast cancer prognosis and treatment.

Giuseppe Sconocchia, Giulia Lanzilli, Valeriana Cesarini, Domenico Alessandro Silvestris, Katayoun Rezvani, Roberto Arriga, Sara Caratelli, Ken Chen, Jinzhuang Dou, Carlo Cenciarelli, Gabriele Toietta, Silvia Baldari, Tommaso Sconocchia, Francesca De Paolis, Anna Aureli, Giandomenica Iezzi, Maria Antonietta Irno Consalvo Consalvo, Francesco Buccisano, Ilaria Del Principe, Luca Maurillo, Adriano Venditti, Alessio Ottaviani, and Giulio Cesare Spagnoli

DOI: <https://doi.org/10.26508/lsa.202201590>

Corresponding author(s): Giuseppe Sconocchia, Istituto di Farmacologia Traslazionale

Review Timeline:	Submission Date:	2022-07-04
	Editorial Decision:	2022-08-26
	Revision Received:	2022-09-02
	Editorial Decision:	2022-09-05
	Revision Received:	2022-09-07
	Accepted:	2022-09-07

Scientific Editor: Novella Guidi

Transaction Report:

August 26, 2022

Re: Life Science Alliance manuscript #LSA-2022-01590-T

Dr. Giuseppe Sconocchia
CNR Institute of Translational Pharmacology (IFT), Rome Italy
Biomedicine
Via Fosso del Cavaliere 100
Rome, RM 00133

Dear Dr. Sconocchia,

Thank you for submitting your manuscript entitled "Antibody-independent antitumor effects of CD32-chimeric receptor T cells: implications for breast cancer prognosis and treatment." to Life Science Alliance. The manuscript was assessed by an expert reviewer, whose comments are appended to this letter. We invite you to submit a revised manuscript addressing the Reviewer comments.

When submitting the revision, please include a letter addressing the reviewer comments point by point.

Thank you for this interesting contribution to Life Science Alliance. We are looking forward to receiving your revised manuscript.

Sincerely,

B. MANUSCRIPT ORGANIZATION AND FORMATTING:

Reviewer #1 (Comments to the Authors (Required)):

It has been shown that the Fc receptor CD16 can bind to surface ligands in addition to immunoglobulins, triggering the activation of NK cells and ADCC. In this novel and interesting paper, Sconocchia and coll, used CD32-CAR T cells that they previously generated to identify putative surface ligands recognized by the Fc receptor CD32 on tumor cells. They examined 20 distinct cell types of which 15 were cancer cell lines and demonstrated that that CD32-CAR T cells recognize putative ligands on two triple negative cancer cell lines (MDA-MB-468 and MDA-MB-23), as well as other tumor cell lines, including HER2+ HCC-1954 cells, and CRC HT29. Importantly, CD32-CAR T cells did not recognize normal cell lines, such as primary fibroblasts and a myoblast cell line. The authors further performed differential gene expression analysis between cells positive and negative for CD32-CAR T cell recognition, identifying a number of potential candidate molecules. Finally, they showed that target cell expression of ICAM1 is essential for CD32-CAR T cells recognition.

This is a brilliant application of a novel biosensor for the identification of CD32 unknown ligands. Moreover, this work has also led to demonstrate a role of CD32-CAR mediated cytotoxicity independent from ADCC which may be important for therapeutic avenues of intervention. The experiments are extensive and convincing. The paper is well-organized and written. I have only minor suggestions that the authors may consider

1. In introducing Fc Receptors it is unclear whether the authors are describing human or mouse Fc receptors. The authors should clarify this from the beginning
2. In the beginning of the results, the authors say that they purified T cells for scRNA seq. It is unclear from where these cells were purified. From the xenograft mouse model in which the cells were transferred? From in vitro cultures with tumor cells?

Concern 1: in introducing Fc Receptors, it is unclear whether the authors are describing human or mouse Fc receptors. The authors should clarify this from the beginning

Author's reply: we have clarified the human nature of the Fc-CR utilized in our study. Please, in find the addition,. Changes are indicated in red and are located in the abstract and introduction

Concern 2: in the beginning of the results, the authors say that they purified T cells for scRNA seq. It is unclear from where these cells were purified. From the xenograft mouse model in which the cells were transferred? From in vitro cultures with tumor cells?

Author's reply: a clarification sentence has been added to the beginning of the first paragraph of the result section in page 7.

Based on the guide lines of LSA policy, we had to shorten the title down to 100 characters without changing the meaning.

September 5, 2022

RE: Life Science Alliance Manuscript #LSA-2022-01590-TR

Dr. Giuseppe Sconocchia
Istituto di Farmacologia Traslazionale
Biomedicine
Via Fosso del Cavaliere 100
Rome, RM 00133
Italy

Dear Dr. Sconocchia,

Thank you for submitting your revised manuscript entitled "Direct CD32 T cell cytotoxicity: implications for breast cancer prognosis and treatment.". We would be happy to publish your paper in Life Science Alliance pending final revisions necessary to meet our formatting guidelines.

- please add a category for your manuscript to our system
- please add a callout for Figure 4F and Figure 7 to your main manuscript text
- please remove the panel A from the figure 7 legend; since this is the only panel, you do not need to label the panel individually

A. FINAL FILES:

B. MANUSCRIPT ORGANIZATION AND FORMATTING:

**Submission of a paper that does not conform to Life Science Alliance guidelines will delay the acceptance of your

manuscript.**

The license to publish form must be signed before your manuscript can be sent to production. A link to the electronic license to publish form will be sent to the corresponding author only. Please take a moment to check your funder requirements.

Sincerely,

September 7, 2022

RE: Life Science Alliance Manuscript #LSA-2022-01590-TRR

Dr. Giuseppe Sconocchia
Istituto di Farmacologia Traslazionale
Biomedicine
Via Fosso del Cavaliere 100
Rome, RM 00133
Italy

Dear Dr. Sconocchia,

Thank you for submitting your Research Article entitled "Direct CD32 T cell cytotoxicity: implications for breast cancer prognosis and treatment.". It is a pleasure to let you know that your manuscript is now accepted for publication in Life Science Alliance. Congratulations on this interesting work.

DISTRIBUTION OF MATERIALS:

Again, congratulations on a very nice paper. I hope you found the review process to be constructive and are pleased with how the manuscript was handled editorially. We look forward to future exciting submissions from your lab.

Sincerely,
